# Assessment of the Orthodontic External Resorption in Periodontal Breakdown—A Finite Elements Analysis (Part I)

**DOI:** 10.3390/healthcare11101447

**Published:** 2023-05-16

**Authors:** Radu Andrei Moga, Cristian Doru Olteanu, Mircea Daniel Botez, Stefan Marius Buru

**Affiliations:** 1Department of Cariology, Endodontics and Oral Pathology, School of Dental Medicine, University of Medicine and Pharmacy Iuliu Hatieganu, Str. Motilor 33, 400001 Cluj-Napoca, Romania; 2Department of Orthodontics, School of Dental Medicine, University of Medicine and Pharmacy Iuliu Hatieganu, Str. Avram Iancu 31, 400083 Cluj-Napoca, Romania; olteanu.cristian@umfcluj.ro; 3Department of Structural Mechanics, School of Civil Engineering, Technical University of Cluj-Napoca, Str. Memorandumului 28, 400114 Cluj-Napoca, Romania; mircea.botez@mecon.utcluj.ro (M.D.B.); marius.buru@mecon.utcluj.ro (S.M.B.)

**Keywords:** external root resorption risks, periodontal breakdown, orthodontic movements, finite elements analysis, failure criteria

## Abstract

This Finite Elements Analysis (FEA) assessed the accuracy of Tresca failure criteria (maximum shear stress) for the study of external root resorption. Additionally, the tooth absorption–dissipation ability was assessed. Overall, 81 models of the second mandibular premolar, out of a total of 324 simulations, were involved. Five orthodontic movements (intrusion, extrusion, rotation, translation, and tipping) were simulated under 0.6 N and 1.2 N in a horizontal progressive periodontal breakdown simulation of 0–8 mm. In all simulations, Tresca criteria accurately displayed the localized areas of maximum stress prone to external resorption risks, seeming to be adequate for the study of the resorptive process. The localized areas were better displayed in the radicular dentine–cementum component than in the entire tooth structure. The rotation and translation seem prone to a higher risk of external root resorption after 4 mm of loss. The resorptive risks seem to increase along with the progression of periodontal breakdown if the same amount of applied force is guarded. The localized resorption-prone areas follow the progression of bone loss. The two light forces displayed similar extensions of maximum stress areas. The stress displayed in the coronal dentine decreases along with the progression of bone loss. The absorption–dissipation ability of the tooth is about 87.99–97.99% of the stress.

## 1. Introduction

Root resorption is a common, hard to anticipate side-effect of orthodontic treatment, with a reported variable prevalence (e.g., 20–100% [1,2,3]) affecting the dentine and cementum of the root and crown [1,4,5]. Most root resorptions are limited to small areas/lacunae found on the root surface, clinically insignificant, and appearing after 10–35 days after the continuous force was applied [6,7]. The mechano-biology of the process is not entirely understood, and is difficult to clinically predict, being considered of iatrogenic etiology [3]. In permanent teeth, the orthodontic root resorption occurs both on the external side of the tooth (external root resorption) and within the root canal and pulp chamber (internal resorption) [1,5].

The external root resorption is an unavoidable, self-limiting pathological irreversible loss of dentin and/or cementum, occurring in three phases (i.e., initiation, resorption, and repair) [5]. The resorptive process could interest the external root surface (a non-infectious, self-limiting and pressure-induced resorption, and with a prevalence of 1–5%), and/or cervical third (a cervical resorption in any direction and with variable extent, bellow the epithelial attachment, and with a prevalence of 0.02–2.3%) [5]. 

When orthodontically triggered, the severity is usually influenced by the magnitude of force (e.g., 0.5–1 N/approx. 50–100 gf are considered to be optimal and safe), by the amount of apical displacement and the direction of force (e.g., reports of intrusion to be more prone to resorption), by the root morphology and length, and by the time of appliance (i.e., the increased time is more prone to risks) [1,5,8,9,10,11,12,13]. Nevertheless, the optimal amount of force is still a subject of controversy, due to many reports of various amounts of force 0.28–3.31 N [14,15,16]. There are no available studies about the issue of orthodontic re-treatments and the radicular resorption risks. 

The pathogenesis is attributed to damage to the pre-cementum due to pressure determined by the orthodontic forces [5]. In the periodontal ligament (PDL) the orthodontic force produces compression in blood vessels and circulatory disturbances (altering the physiological maximum hydrostatic pressure/MHP) triggering the resorptive processes until the pressure source is removed [5]. The root resorption is usually asymptomatic (small cavities/lacunae of approx. 6 µm after the first 10 days of force appliance [7]), with normal pulp response to the vitality tests and without any signs of endodontic sufferance [5]. If the resorptive lesion is identified, a temporary pause of around 3 months usually allows the resorptive surface cementum to repair [5]. Nonetheless, if the dentin layer is compromised (after the first 15–25 days of force appliance [7,17]) the destruction process is irreversible [3]. The cervical resorptive defects are usually irreversible and restored using various filling materials [5]. Nevertheless, the individual response to orthodontic forces depends more on individual susceptibility, genetics, and ethnicity than gender and age [17].

The excessive orthodontic force is responsible for external root resorption (the higher the force, the faster the resorption lacunae appear) [3,4]. Nonetheless, a small/light orthodontic force seems not to effectively stimulate the alveolar bone remodeling process (PDL considered the initiating factor), with reduced orthodontic movement and extended treatment time, but without resorptive risks [4]. Nevertheless, due to the high vascularization of PDL, and increased risks of serious circulatory disturbances that might lead to pulp and neuro-vascular bundle (NVB) ischemia and further PDL and tissue loss, the maximum hydrostatic physiological pressure (MHP) of 12.8–16 KPa (i.e., 80% of systolic pressure) should not be exceeded, while the lower limit of 4.7 KPa (in order to initiate the movement), is recommended to be exceeded [3,4,8,9,10,11,18].

The amount of force to be applied over the tooth (i.e., light forces could also face increased root resorption if the application time is increased [2,19]), the modality of absorption–dissipation of stress by the tooth and surrounding support structures, and the amount of stress that reaches the PDL, still remain subjects of controversy [7,10,11]. Depending on the type of movement, the high points of pressure where the force is high are more prone to resorptive processes [19], continuing for another 4 weeks after the force action stopped [2]. There are reports of intrusion to be prone to apical root resorption, extrusion to provoke cervical third resorption, and for the rotation the lacunae to be prevalent in the middle third of the root [19]. The reparative process of root resorption starts in periodontium when the orthodontic force becomes discontinued and/or reduced below a certain level [2]. 

When studying the resorptive processes concerning the root, it is important to understand the anatomical micro-architecture and physical properties of each of the components of the investigated region (i.e., dentine, enamel, cementum, dental pulp–NVB, PDL, bone) [13,20,21,22]. A large amount of the stress determined by the orthodontic forces is absorbed, dissipated, and reduced by the tooth components (i.e., dentine and cementum), so that only a small part of the applied force (i.e., stress) reaches the PDL, dental pulp–NVB, and bone [8,12,18,20,21,22,23,24]. Biomechanically, the tooth structure, PDL, and dental pulp–NVB are considered to resemble a ductile material (with a certain brittle flow mode) [8,12,18,20,21,22,23,24].

The study of the resorptive process of the root is performed by in vivo–in vitro and FEA studies, and each method has benefits and limits. In vivo–in vitro studies [7,17] can directly observe the consequences of loads and movements over the entire structures (i.e., a global view) but without individually analyzing each anatomical component and the amount of stress, or without being able to change the experimental conditions [20]. It has been reported that in in vitro simulations the anatomical tissues suffer a variable change of physical properties [20], and that FEA is preferable due to serious discrepancies related to design and validation of in vitro results [25]. FEA individually assesses each component and enables the change of experimental conditions according to the observations, but as a major limit, cannot perfectly reproduce the anatomical and biomechanical conditions, being accurate only if the adequate failure criteria is employed and the input data are correct [8,9,12]. 

In the engineering field, FEA is extremely accurate and widely used for both testing and development, while in the dental medicine field, due to a misunderstanding of the yielding of materials theory (which is behind the FEA functioning principles), the studies provided contradictory results insufficiently corelated with the clinical data [8,9,12]. Recent FEA dental studies reported that if the material-based indications of the failure criteria employment are followed and correlated with correct input data, the FEA method can provide accurate results that are validated by correlations with the clinical data [8,9,10,11,12,18]. Thus, using FEA means, 0.6–1.2 N has been reported to be the maximum amount of force safely applied in 0–8 mm of the horizontal periodontal breakdown process for the PDL and the dental pulp–NVB complex, correlating with the MHP limit (PDL stress remained lower than MHP) and the clinical available data [8,9,10,11,12,18]. By comparing five failure criteria, the ductile behavior of the tooth structure, of PDL, and of dental pulp–NVB (the von Mises and Tresca criteria were reported as adequate, with Tresca being considered more suitable) have been reported [8,9,10,11,12,18].

Few studies are available regarding the biomechanical behavior of tooth and surrounding support tissues when subjected to small orthodontic forces during the periodontal breakdown process. Additionally, no data related to the risks of the external orthodontic root resorption process have been found, even though various levels of bone loss are usually present in orthodontic patients.

There are multiple FEA studies of the PDL (the initiating factor in orthodontic movement) that approaches the subject of resorption, employing failure criteria adequate either for brittle materials (the main difference between brittle (maximum principal S1 and minimum principal S3 stress [26,27,28,29,30]) and ductile (von Mises and Tresca stress [28,29,30]) is related to the way the material breaks/deforms under loads, and is called yielding theory), or gas/liquids (maximum hydrostatic pressure [3,4,14,16,28,31,32]). Nonetheless, these reports did not consider assessing the correlation between their results and the anatomical inner microstructure of the tissues or their biomechanical response to forces (providing quantitative and qualitative reports debatable when compared to clinical data [8,9,10,11]), despite the proven ductile resemblance [8,9,10,11,12,20,23,24] of dental tissues. 

In vivo studies reported that resorption is prone to appear on the compression/pression side and only rarely on the tension side [7]. The magnitude of applied force correlation with the severity of root resorption (extension, depth, diameter) is still a subject of controversy with reported pros [17] and cons [7]. Nonetheless, the higher the force, the greater the volumetric loss seems to be [3,7,17]. It has been suggested that the root resorption is correlated more with the stress distribution than the force magnitude (lower volumes of resorptive lesions are associated with bodily movements than tipping due to different stress distribution, while intrusion seems to produce higher force per area producing increased tissue necrosis) [17]. Thus, it reinforces the concept that lesion location is dependent on the stress concentration on the surface of the root, which is indirectly determined by the direction of applied force and root anatomical morphology [17]. Thus, the accuracy of the FEA is dependent on the 3D reconstruction of the anatomical curvature of the tooth surface [17]. 

FEA is the only available method of study for root resorption allowing the individual assessment of stress for each component. Nonetheless, the current FEA regarding the root external resorption in intact periodontium employed mostly the hydrostatic pressure failure criteria [3,7,17,31,32]. This is not adequate for analyzing the ductile materials (such as the tooth structure [8,12,20,23,24]), with inaccurate results regarding the areas of high stress (e.g., entire root apical third for intrusion [32], or cervical and apical third for tipping [3,31]) when compared with in vivo–in vitro experiments (i.e., localized small lacunar areas). However, by employing the Tresca failure criteria (reported more adequate [12]), FEA could be able to accurately localize and display the maximum/high-stress areas and to make the external root resorption more predictable.

This research is the first of this type to study the correlations of the external root resorption and the horizontal periodontal breakdown, employing an adequate failure criterion (suited and documented for these tissues) which supplies correct results. Earlier FEA studies failed in supplying exact location of areas susceptible to resorptive processes by not using the adequate failure criteria.

The aims of the present simulation were as follows: (a) an assessment using Tresca criteria that is adequate for the study of root resorption, (b) the assessment of the areas more prone to orthodontic external root resorption when 0.6 N/60 gf and 1.2 N/120 gf are applied during the 0–8 mm periodontal breakdown process under multiple orthodontic forces, (c) the ability of dental structures to absorb and dissipate the two applied forces.

## 2. Materials and Methods

The present finite elements simulation is a part of a larger study (with clinical protocol nr.158/02.04.2018) aimed at studying the biomechanical behavior of tooth and surrounding periodontal tissues during the horizontal periodontal breakdown [8,9,10,11,12]. Each of the simulations of this research derives from the results and conclusions of the previous one, thus enabling a more complex picture of the biomechanical processes occurring in these tissues during orthodontic movements. The objective of the present simulation was to display the areas more prone to external root resorptive risks for each of the five most used orthodontic movements.

The present simulation was conducted over 81 3D models of the second lower premolar with various levels of surrounding support tissues (a total of 324 FEA simulations), acquired by radiological means from 9 patients with the mean age of 29.81 ± 1.45 years. There were 4 males and 5 females, and all had oral informed consent.

The initial number of examined patients was larger, but only nine of them qualified due to the restrictive inclusion criteria (i.e., various levels of periodontal breakdown mainly at the cervical third root level, non-inflamed periodontium, no mandibular tooth loss, intact teeth, no malposition, orthodontic treatment indication, willingness for regular monitoring). The mandibular premolars were selected both out of a desire to study the area with higher loadings and periodontal problems and because of the finite element studies available in the literature on which to perform data comparison and validation.

The sample size of 9 (the 324 FEA simulations were conducted over 81 3D models) was acceptable due to the fact that the current FEA available studies usually used a sample size of 1 (one model acquired from one patient then subjected to several simulations) due to difficulties in building the 3D models.

The initial information on which our 3D models were based came from radiological analysis performed for diagnostic and control purposes using CBCT (cone beam computed tomography) examination (of the mandibular arch with the two premolars and molars), with a voxel size of 0.075 mm, and FOV of 50 × 50 mm (ProMax 3DS, Planmeca, Helsinki, Finland).

A single experienced clinician performed the manual image reconstruction process (ensuring higher anatomical accuracy compared to the automatic process) using Amira 5.4.0. (Visage Imaging Inc., Andover, MA, USA) reconstruction software. The tooth’s components (enamel, dentin–cementum, dental pulp, neuro-vascular bundle) and surrounding support tissues (periodontal ligament, cortical and trabecular bone) were selected on each DICOM slice (Hounsfield grey units) and assembled in a 3D model (nine original models of each of the nine patients with various levels on bone loss at the cervical third of root and containing only the second lower premolar, e.g., Figure 1 and Figure 2). In each of these nine models only the missing surrounding support tissues (PDL and bone) were reconstructed as close as possible to the original anatomical shape. Thus, 9 models with intact periodontal support were obtained (5.06–6.05 million C3D4 tetrahedral elements, 0.97–1.07 million nodes, global element size of 0.08–0.116 mm, e.g., Figure 1 and Figure 2). 

Each of these models was then subjected to a horizontal bone and PDL reduction of 1 mm (from 0–8 mm of tissue loss) simulating a progressive horizonal periodontal breakdown process, obtaining a total of 81 models (9 models for each of the intact periodontium models).

The manual image reconstruction process, despite supplying a better anatomical accuracy to the 3D models, has a limit to the presence of surface anomalies and irregularities (as to be expected). The internal control algorithms do not allow the later steps in model reconstruction and the finite elements analysis unless these anomalies and irregularities are placed in non-essential areas (i.e., the stress display areas must be quasi-continuous). The models herein, after several smoothing processes, presented surface non-essential anomalies and irregularities, while the stress display areas were quasi-continuous. Moreover, the mesh testing process reported no errors and only a limited number of surface element warnings (e.g., for the models displayed in Figure 1 and Figure 2, 264 element warnings (representing 0.0043%) for the entire model of 6.05 million C3D4 elements, 63 element warnings (0.00677%) for the 930023 elements of the tooth, bracket and PDL, and 39 element warnings (0.00586%) for the 665,501 elements of the tooth and bracket). Thus, despite a certain number of irregularities that remained after the smoothing processes, the extremely small number of element warnings (e.g., a global 0.0043% for Figure 1 and Figure 2) did not interfere with the quality and accuracy of the simulations and results.

The periodontal ligament thickness varied between 0.15–0.22 mm and included the neurovascular bundle of the dental pulp for each of the 81 models. Because of the similar physical properties and difficulties that needed to be separated on the DICOM slices, the cementum was reconstructed as dentine (Table 1). The assumed boundary conditions were the perfectly bonded interfaces, isotropy, linear-elasticity, and homogeneity, since under small loads of around 1 N and extremely reduced movements all tissue shows linear elasticity. Additionally, almost all FEA studies regarding this subject used similar boundary conditions. 

The finite elements simulations (totaling 324) were performed using ABAQUS 6.13-1 software (Dassault Systèmes Simulia Corp., Maastricht, The Netherlands), employing the Tresca failure criteria (the maximum shear stress criterion) more adequate for the study of dental tissues than other failure criteria [8,9,12]. The orthodontic movements simulated were the extrusion, intrusion, rotation, translation, and tipping under 2 applied forces of 0.6 N/approx. 60 gf and 1.2 N/approx. 120 gf at the bracket level (e.g., Figure 1). The 0.6 N and 1.2 N were selected based on the results and conclusions of 2 previous studies of our group regarding the maximum safely applied force in periodontal ligament and dental pulp and neuro-vascular bundle [10,11] and for being able to perform correlations with the PDL and dental pulp studies.

The results of the simulations were both qualitative (the color-coded projections of the shear stress display in both dentin and entire tooth structure, e.g., Figure 3, Figure 4, Figure 5, Figure 6, Figure 7 and Figure 8) and quantitative, determining the maximum average shear stress values for the apical, middle, and cervical thirds of the root, and the crown (Table 2). Both qualitative and quantitative results were corelated with the results of a previous study [11] (the shear stress display in PDL under 0.6 and 1.2 N in the periodontal breakdown process) to be able to evaluate the absorption–dissipation ability of the dental tissues (Table 3).

The color-coded projections of the shear stress were displayed in both tooth structure as well as in the dentine components as red-orange maximum (prone to high resorptive risks), yellow and yellow-green (moderate resorptive risks) and blue-green (reduced resorptive risks).

The two analyzed structures were the entire tooth structure (including the dentine–cementum, enamel, and bracket) and the radicular dentin–cementum and coronal dentine components to highlight the areas more prone to external root resorption risks.

## 3. Results

This study was performed over 81 3D models of the second lower premolar (from 9 patients) totaling a number of 324 FEA simulations. No differences associated with age, gender, or periodontal status were detected. The FEA quantitative and qualitative results were displayed in Figure 3, Figure 4, Figure 5, Figure 6, Figure 7 and Figure 8 and Table 2 and Table 3.

In each of the analyzed two structures the maximum stress was displayed. At the level of the tooth, this occurs in certain areas due to the biomechanical behavior of the component tissues. Meanwhile, in radicular dentine–cementum and coronal dentine (due to the absence of interferences determined by the other anatomical components) the maximum shear stress (the same stress but quantitatively smaller) was displayed more accurately and concisely in localized potential lacunae.

Qualitatively, the Tresca stress manifested in the entire tooth structure, radicular dentine–cementum, and coronal dentine components, and is displayed as various color-coded projections (e.g., Figure 3, Figure 4, Figure 5, Figure 6 and Figure 7 for 0.6 N/60 gf and Figure 8 for 1.2 N/120 gf), showing small, localized areas/lacunae of maximum stress (red-orange) surrounded by high-stress (yellow) areas, all prone to resorptive risks. Quantitatively (Table 2), the stress in the radicular dentine–cementum component of the tooth was lower than the displayed stress for the entire tooth structure, for all movements and bone levels (0–8 mm of loss). Nonetheless, as the bone loss process progressed to the maximum (prone to the resorptive process), quantitative shear stress values increased, so that at 8 mm bone loss the stress value in the dentinal component was the same as that which manifested in the entire tooth structure (see Table 2). Biomechanically, once the periodontal support is reduced, the same forces cause a stress increase by more than ten times at root level (i.e., apical, and middle third for translation, rotation, and tipping) vs. quadrupling (rotation) and doubling (translation and tipping) for the coronal part. Nevertheless, the amount of stress at both 0.6 N and 1.2 N, were lower than the reported maximum shear stress (29–73.1 MPa dentin, 53.9–104 MPa enamel–dentin [21]). 

### 3.1. Intrusion and Extrusion

In the intrusion and extrusion movements (e.g., Figure 3 and Figure 4) in the entire tooth structure, in intact periodontium the stress was concentrated in the vestibular cervical third while the periodontal loss progressed and extended to the entire root, both on the vestibular and lingual sides. When analyzing only the radicular dentine–cementum component, the maximum quantitative stress was less than that for the entire tooth structure (as biomechanically expected to be), but qualitatively the stress was displayed in the entire root over the entire surface with a maximum (limited color-coded red-orange areas) located at the periodontal support level on vestibular side. The color-coded red-orange areas are those more prone to suffering from the external orthodontic root resorption process, seeming to be more visible around 4 mm of bone loss (while at 0 and 8 mm of loss they are less visible). The orange color-coded areas (high stress) could also be prone to moderate resorptive process and are visible on the vestibular root side at all bone levels including the apical third (at 8 mm of loss). The coronal dentine component showed the highest stress area (small and colored in green) under the bracket position in intact periodontium, while during the periodontal loss, the displayed stress showed a progressive decrease.

### 3.2. Rotation

The rotational movement (e.g., Figure 5) displayed in the tooth structure reduced quantitative cervical stress for intact periodontium, but also showed a visible increase in stress in the entire root surface for the reduced periodontium, especially after 4 mm of loss (with visible limited yellow coded areas). In the radicular dentine–cementum component, the maximum shear stress color-coded red-orange areas (prone to root resorption) were displayed on the root medial and distal sides at periodontal support levels (with limited areas of orange on the lingual side of the root after 4 mm of loss). The yellow-coded areas (high shear stress, also prone to resorptive processes) were more extended on the surface as the maximum stress areas and were present on all four sides of the root (with higher extension after 4 mm of bone loss). In the apical and middle third of the root, the quantitative amount of shear stress quadruples after 3–4 mm of tissue loss, reaching a 7–9 times stress increase when compared with intact periodontium. The coronal dentine component showed a maximum stress area (prone to resorption) under the bracket position in intact periodontium, while during the periodontal loss, the displayed stress showed a progressive decrease.

### 3.3. Tipping

The tipping (e.g., Figure 6) in intact periodontium displayed a cervical lingual side barely visible area of stress for the tooth structure, but with a significative increase in stress as the bone loss process progressed. The maximum shear stress (red-orange color) was displayed on the lingual side of the root at 8 mm of loss. In the radicular dentine–cementum component at 0 mm bone loss, the stress areas were visible on the entire lingual side and vestibular cervical third of the root. The maximum stress (red-orange color-coded areas) was present from 3 to 4 mm of periodontal breakdown at the periodontal support level both on the vestibular and lingual sides (with resorptive process more prone to appear lingually). High-stress areas (yellow color-coded, also prone to resorptive risks) were visible on the entire vestibular and lingual surface of the root. In the apical and middle third of the root, as the periodontal breakdown process progressed, the quantitative stress tripled after 4 mm of loss and reached 6.5–8.6 times (at 8 mm) higher than the intact periodontium stress. The coronal dentine component displayed a progressive decrease in stress during the progression of bone loss. 

### 3.4. Translation

In intact periodontium, the translation movement (e.g., Figure 7) displayed lingual, mesial, and distal side cervical stress in the tooth structure. After 4 mm of tissue loss, visible maximum shear stress areas (red-orange surrounded by extended yellow color-coded areas) prone to resorptive processes were displayed on the lingual and distal sides of the root at the periodontal support level. In the radicular dentine–cementum component, the maximum stress red-orange areas (more prone to resorption) were visible on both intact and reduced periodontium on the mesial, lingual, and distal sides of the root at the periodontal support level. The high-stress yellow areas (with a lesser risk of resorptive process) surrounded the red-orange areas and were of higher extent after 4 mm of tissue loss. The apical and middle third of the root showed the highest stress increase up to 11 times at 8 mm of loss (5 times at 4 mm and 11 times at 8 mm for the middle third, and 1.6–8.6 times after 4 mm of loss for the apical third). The coronal dentine component displayed a progressive stress decrease during the 0-8 mm bone loss simulation.

Both the 0.6 N and 1.2 N applied forces displayed similar qualitative results with almost similar surface extension (e.g., Figure 3, Figure 4, Figure 5, Figure 6 and Figure 7 vs. Figure 8), and a doubling of the quantitative values for the 120 gf force compared with the 60 gf force.

The display of the color-coded red-orange (higher resorptive risks) and yellow (lesser resorptive risks) small, localized areas follow the periodontal breakdown progress, with a significant increase after 4 mm of bone loss. 

Thus, the more periodontal support is reduced, the more visible are the red-orange maximum shear stress and yellow high shear stress small, localized areas and thus the risk of localized orthodontic external root resorption increases.

In the coronal dentine component, the rotation displayed a maximum red-orange stress under the bracket in intact periodontium, signaling a potential resorptive area (higher stress also for the other movements), followed by a progressive stress decrease during the bone loss (for all five movements).

When correlating the quantitative results (Table 2) with those reported by an earlier study of our group with focus on the periodontal ligament (Table 3) [11], a more detailed overview of biomechanical process of absorption and dissipation of the stress in the tooth structure and the surrounding periodontium appears. Thus, the absorption–dissipation ability of the tooth structure (due to the dentine–cementum physical properties) is qualitatively displayed in Figure 3, Figure 4, Figure 5, Figure 6, Figure 7 and Figure 8 and quantitatively in Table 2 and Table 3. For 0.6 N and 1.2 N of force, the tooth structure absorbs and dissipates 87.99–97.99% of the stress before reaching the periodontal ligament, while the PDL stress (lower that the reported MHP) represents 0.74–4.08% in apical, 2.01–5.84% in middle, and 4.77–10.24% in the cervical third from the total stress displayed in the tooth structure (for all movements and periodontal breakdown process). Therefore, starting from these correlations, from 0.6 N/60 gf applied over the tooth, only a maximum of 0.024 N/2.4 gf in apical, 0.035 N/3.5 gf in middle, and 0.061 N/6.1 gf in the cervical third PDL arrived during the periodontal breakdown progress when the 5 orthodontic forces were applied. For a higher force of 1.2 N/120 gf, the PDL was subjected to a maximum of 0.0489 N/4.9 gf apically, 0.07 N/7 gf in the middle, and 0.1228 N/12.3 gf cervically, throughout the bone resorption process of 0–8 mm.

Both quantitatively (Table 2) and qualitatively (e.g., Figure 3, Figure 4, Figure 5, Figure 6, Figure 7 and Figure 8), the rotation and translation (closely followed by tipping) were the movements more prone to produce external orthodontic root resorption localized areas (red-orange and yellow color-coded), especially after 3–4 mm of tissue loss in both tooth structure and dentine–cementum component. The red-orange prone to resorption areas seemed to be displayed more around the cervical limit of surrounding supporting tissue levels. The intrusion and extrusion movements seemed less prone to determine resorptive processes. Thus, the periodontal breakdown process enhances the risks of orthodontic external root resorption if the applied force is kept as for the intact periodontium. Nonetheless, in intact periodontium up to 1.2 N/120 gf, the risk of the resorptive process is reduced.

## 4. Discussion

This FEA showed that by using an adequate failure criterion (i.e., Tresca, maximum shear stress) selected according to the material type’s biomechanical behavior (i.e., ductile resemblance with a certain brittle flow mode of dental tissues [8,9,10,11,12,18,20,23,24]), the finite elements analysis was able to display accurately the small, localized areas prone to external root resorption in both intact and reduced periodontium opposingly to other failure criteria [3,31,32] (such as hydrostatic pressure displaying extended areas of maximum stress).

This analysis employed 0.6 N and 1.2 N of orthodontic forces during 5 orthodontic movements in a horizontal periodontal breakdown of 0-8 mm while assessing the absorption–dissipation ability of the tooth as a single stand structure [12]. 

The two forces were selected based on the results of previous FEA [8,9,10,11,18] regarding the maximum tolerable amount of force for PDL and dental pulp and its NVB during the 0–8 mm horizontal periodontal breakdown simulation, which should not exceed the MHP in order to reduce the risks of ischemia, pulp necrosis, and further periodontal tissue loss.

For more accurate results, the FEA simulations (totaling a number of 324) were performed over 81 models from 9 patients.

The maximum shear stress (Tresca failure criteria) was assessed in both the entire tooth structure (having dentine–cementum, enamel, dental pulp, and bracket) and individually in the radicular dentine–cementum and coronal dentine components. The reason for this is related to the fact that the maximum stress was accurately displayed in the form of localized islands/lacunae (red-orange and yellow color-coded) without any biomechanical interference from the other tooth components only in the individual FEA of the radicular dentin–cementum component. However, when analyzing the tooth structure regarding the absorption–dissipation ability of the components, the stressed areas were displayed where the maximum stress occurs in the first place (i.e., usually the enamel under and around the bracket area for no bone loss). Each time the amount of stress displayed in the radicular dentine–cementum component was lower than that of the tooth structure (i.e., being a part of it), as it biomechanically should be. 

The biomechanical behavior of the tooth in intact surrounded support tissues caused maximum stress to occur in enamel around the bracket area for all movements and the two forces. However, with the progression of the periodontal support reduction process, the biomechanical behavior changed so that the maximum stress was additionally displayed in the cervical (4 mm loss) and middle third of the dentin–cementum of the root (8 mm loss), where the bone level is positioned, e.g., Figure 3, Figure 4, Figure 5, Figure 6, Figure 7 and Figure 8. 

The surface extension of the stress areas was almost similar for the two forces in all simulations, since both forces could be seen as light forces that biomechanically produce less displacement than higher forces. Qualitative higher shear stress was visible in the coronal dentine component of the intact periodontium, with a qualitative progressive decrease along with the bone loss, due to changes in the biomechanical behavior during the reduction of tooth periodontal support.

Some of the previous studies [3,7,17,31,32] regarding the external root resorption process had an in vivo–in vitro experimental part assessing the location and depth of resorptive lacunae (teeth subjected to various orthodontic movements and forces for 4–12 weeks, then extracted, and 3D reconstructed and examined). In some of these studies, the in vivo–in vitro part has been completed using a FEA [3,31,32] employing the hydrostatic pressure failure criteria in order assess if the root resorption areas match the clinical lacunae. The reported results [3,31,32] with the hydrostatic pressure criteria were displayed as color-coded projections of the maximum stress areas (prone to resorption) that extended to an entire third or side of the root. They did not accurately match the clinical lacunae and presented extremely high quantitative amounts of stress. Nonetheless, a recent FEA study [12] investigating the adequacy of five different failure criteria (hydrostatic pressure criteria included) for the study of the tooth reported that, both qualitatively and quantitatively, von Mises and Tresca provided more accurate results, with Tresca being more accurate.

Wu et al. [17] performed in vivo–in vitro reports for the rotation movement of the maxillary first premolars for 4 weeks under 25 gf (approx. 0.25 N) and 225 gf (approx. 2.25 N). They found small resorption craters displayed at the boundaries between vestibular/buccal–distal and mesial–lingual sides of the root in all sectors—apical, middle, and cervical third—with more resorption resulting from a heavy force than from a light one. In our FEA simulation (0.6 N and 1.2 N) with Tresca criteria, in intact periodontium we found maximum stress areas (red-orange) for the dentine–cementum component displayed distal–lingual and lingually at furcation level and high-stress areas (yellow) distal-lingual, lingual, mesial, and vestibular sides in cervical third. The maximum stresses in all three sectors of the root (on distal–lingual, lingual, mesial–lingual, and vestibular sides) are displayed progressively along with the increase in the periodontal breakdown process (e.g., Figure 5). It must be pointed out that in their study, Wu et al. [17] applied a higher couple of forces both buccally and lingually (two brackets, each placed on one of the two sides of the first maxillary premolar), which explains the higher root damage. Whereas in our study, the forces were applied only at the bracket level (one bracket on vestibular/buccal side of the second mandibular premolar) and the applied force was much lower. 

Hohmann et al. [32] in the in vivo–in vitro experiment of a mixed study analyzing the intrusion of maxillary first premolars for 4 weeks with no mention of periodontal status, and under 0.5 N (approx. 50 gf) and 1 N (approx. 100 gf), reported limited areas of resorption craters in the apical third of the root and furcation region. The FEA [32] of the same premolar model (195,881–215,887 PDL elements and 71,114–74,777 tooth elements), under the same forces, employing hydrostatic pressure failure criteria displayed extremely high-stress areas (color-coded in grey and with a reported maximum stress of 9.95 TPa) unusually involving the entire apical third sector of the root (not matching the clinical report), suggesting that the entire two rooted apex is resorbable. The FEA intrusion simulation in our study displayed in the dentine–cementum component localized areas/lacunae of maximum shear stress (73.5–109 KPa for 0.6 N, Table 2) in the vestibular-mesial sides, furcation, and apex (e.g., Figure 4). Along with the progression of the periodontal breakdown process, the maximum and high shear stress areas were visible on the vestibular sides (cervical and middle third), and vestibular and lingual apical third. The differences between the clinical report [32] and our simulation originated from the different analyzed teeth (the variable individual response) and the way (and amount of) the forces were applied. The significant differences between the FEA report [32] and ours (e.g., quantitatively: 9.95 TPa maximum apical stress vs. our 73.5 KPa maximum apical stress; and qualitatively: entire apex of tooth area vs. small and isolated localized areas in our simulations), originated mainly from the differences of the employed failure criteria (hydrostatic criteria reported to be biomechanically inadequate for the tooth [12]), but also from different teeth and force appliance. Roscoe et al. [28] in an FEA simulation of resorption risks under 25 gf of the intrusion/buccal tipping of an idealized model of premolar (1.67 mil elements) employing von Mises, S3, and hydrostatic pressure reported a color-coded unusual biomechanical behavior for all three criteria and two movements (seeming to contradict the other FEA color-coded results performed in relatively close circumstances). It was assumed that the difference originated in the applied boundary conditions and the model’s morphology and accuracy.

Chan et al. [7] in an in vivo–in vitro analysis of the tipping force (buccally directed) applied in a maxillary first premolars for 4 weeks, and 25 gf (approx. 0.25 N) and 225 gf (approx. 2.25 N), reported limited lesions on buccal cervical and lingual apical compression surfaces. In another in vivo–in vitro study, Hohmann et al. [31] reported for 4 weeks of lingual torque (tipping) of 3 Nmm (approx. 300 gf) and 6 Nmm (approx. 600 gf) of the maxillary first premolars resorption lacunae only in the apical third (on the lingual side of the lingual and buccal root). Nonetheless, when analyzing the FEA model of the same tooth (152,776–165,254 PDL elements and 56,454–61,541 tooth elements) employing the hydrostatic pressure failure criteria, the extension of the high-stress areas was much larger than those displayed by the in vivo–in vitro experiment, and with an amount of stress of 38.84 KPa [31]. Zhong et al. [3], in their in vivo–in vitro experiment of 12 weeks, applying 25 gf (0.245 N) and 225 gf (2.2 N) buccal tipping force over the mandibular first premolars, reported cervical third buccal side lesions and apical third lingual side (especially along the peri-groove regions). The FEA of the same study [3], employing the hydrostatic pressure failure criteria in the mono rooted premolar model (204,369 C3D10H elements, global size of 1 mm), displayed extended areas of extremely high stress extended in the entire buccal cervical and lingual apical third. It did not accurately match the localized lesions showed by the in vivo–in vitro experiment [3].

Zhong et al.’s [3] study is the only one found to display a comparative image of the effects of the two forces over the analyzed roots, with variable positions, extensions, and depths of the lacunes, variable from one tooth to another and from one force to another, the single visible pattern being their recurrent location on the buccal side and an apical lingual side, in agreement with our results herein. Thus, these studies [3,7,31] seem to support the assumption [1,2,3,4,5,6,7,17] that there is an individual unpredictable reaction to resorption (variable high points of pressure [19]), variable with the patient, tooth, and applied force, and that each tooth should be individually analyzed, which is also in agreement with our analysis.

In our FEA simulations of the tipping in intact periodontium, the high-stress areas were located mainly on the lingual (less on vestibular) side cervical third of the root and in the apical third in the lingual peri-groove regions (a maximum amount of 61.67 KPa), (e.g., Figure 6 and Figure 8). This is comparable with Chan et al. [7], Zhong et al. [3], and Hohmann et al. [31] (only partially) in their clinical reports, that showed that as the periodontal loss progressed, the maximum shear stress areas were more visible, especially on the lingual (less on vestibular) side in the cervical and middle third and the lingual apical peri-groove third. The differences between the in vivo–in vitro studies [3,7,31] and this analysis originated due to the individual variability of tissular response, applied forces, and anatomical differences. However, when corelating the FEA results of the hydrostatic pressure criteria studies [3,31] with our Tresca simulations, the differences are visible due to the diverging employed failure criteria [12], tooth anatomical morphology, and applied forces.

The results of our FEA simulations seem to resemble more those provided by the clinical studies [3,7,17,31,32]. There are visible differences between our herein Tresca criteria simulations and hydrostatic pressure criteria [3,31,32] ones (mainly due to the differences in the biomechanical behavior described by the two failure criteria [8,9,12]).

Field et al. [29] employed von Mises (comparable with Tresca, but Tresca is specially designed for non-homogenous ductile materials, while von Mises is for homogenous ones), hydrostatic pressure, and maximum S1 and minimum S3 principal stress in an FEA study of the tipping movement. The analyses [29] were performed over one 3D mandibular model of 32,812 elements of the incisor–canine–premolar, and one 3D model with only the canine of 23,565 elements. In total, there was 0.35 N of tipping (0.5 N of resulting load), reporting 60-192.6 KPa on the cervical region and the outer surface of the root for von Mises (vs. 61.67–122.61 KPa herein), 0.8–32 KPa of hydrostatic pressure in the apical third of the root (almost double the 16 KPa of the physiological MHP), and S1 tensile and S3 compressive amounts of 235.5–324.5 KPa for the periodontal ligament (almost 14.5 times higher than MHP). Based on the reported excessive amounts of stress, Field et al. [29] suggested an increased risk of periodontal loss and apical root resorption, which is unusual for a force considered to be light (0.35 N). The von Mises color-coded projections of stress in the radicular dentin–cementum [29] displayed extended areas of red-orange maximum stress (of 70 KPa at the crown, cervical, middle, and apical third vestibular, distal, and lingual sides) suggesting also an unusual pattern external root of resorption (but comparable to those displayed by other FEA [3,31,32]). Despite the von Mises quantitative results’ close resemblance to the results herein, the qualitative differences originated mainly in the applied loadings in 3D models with different degrees of anatomical accuracy (e.g., Field et al.’s [29] model was made of 23,565-32,812 elements with a global element size of 1.2 mm vs. a model of 5.06–6.05 million C3D4 tetrahedral elements with a global element size of 0.08–0.116 mm).

The absorption–dissipation ability of the tooth structure is biomechanically acknowledged [8,12,20,23,24]; nevertheless, it has not yet been investigated (no studies were found, except an earlier report [12] of our group). The stress correlation of this study with a previous FEA study of PDL [11] under the same applied forces and boundary conditions allowed for the quantification of the tooth absorption–dissipation capacity, reporting 87.99–97.99% of the stress being absorbed before reaching the periodontal ligament (similar with our previous report of 86.66–97.5% [12], in line with the reports regarding the biomechanical properties of the tooth components [8,20,21,23,24] and clinical data of the optimal amount of applied force [1,5,13].

FEA can supply an accurate display if the failure criteria based on the analyzed material type is employed and the input data are correct [25]. There are significant differences between failure criteria, related to the way the analyzed material biomechanically deforms when subjected to stress. Each failure criterion was specially designed for accurately describing a certain type of biomechanical deformation based on the inner structure of each material (elastic–ductile, plastic–brittle, compressible/incompressible–hydrostatic, etc.).

The dental tissues are considered to resemble to ductile non-homogenous materials (with a certain brittle flow mode) [8,9,10,11,12,20,23,24], being able to sustain elastic and plastic deformation without fracture (before the yielding phase), with von Mises and Tresca criteria being more appropriate. The brittle materials cannot sustain deformation under stress, fracturing when the yielding point is reached (maximum S1 and minimum S3 principal stress). The enamel is proven to be brittle, forming cracks and fractures under loads, and directly transmitting the loads, but it is only a small percentage of the entire tooth volume. All other components, including the bracket, are ductile or of ductile resemblance [20,22,24].

The hydrostatic stress was specially designed for liquids where there is no shear stress. It must be emphasized that the shear stress is a crucial factor in the biomechanical behavior of dental tissues, thus, it must not be neglected. The hydrostatic stress [3,4,14,16,28,31,32] was employed in the study of PDL, using the Ogden hyper elastic model, which is not correct when the anatomical inner structure and the clinical biomechanical behavior are analyzed. This is the explanation for the differences between in vivo–in vitro experiments and hydrostatic reports. Moreover, in hydrostatic FEAs Hohmann et al. [31,32] reported amounts of PDL apical hydrostatic stress that were extremely high (e.g., TPa vs. KPa—the MHP is only of 16 KPa), while Wu et al. [14,15,16] reported 0.28–3.31 N as optimal for intact PDL (for canine, premolar, and lateral incisive). Additionally, this displayed significant differences for the same tooth (e.g., canine: rotation 1.7–2.1 N [16] and 3.31 N [14]; extrusion 0.38–0.4 N [16] and 2.3–2.6 N [15]; premolar: rotation 2.8–2.9 N [14]), higher than the reported optimal clinical interval of 0.5–1.2 N.

Nevertheless, from the biomechanical point of view, under an extremely small amount of force of around 1 N, all failure criteria display some partially comparable results (by describing a similar biomechanical behavior (as reported by [8,9,12] in the comparations of the failure criteria for PDL, pulp–NVB, and tooth), but with the increase in applied force, the differences become more and more visible [30].

The main limit of the FEA method is that could not accurately reproduce the clinical situation, and it needs validation through comparation/correlation with clinical and in vitro data [25]. This study did not benefit from an experimental in vivo–in vitro experimental part (needed for a better validation of the results), thus the comparation with other studies was necessary. However, due to the similar boundary conditions, the results could be correlated. The sample size in our study was nine (nine models, from nine patients, but with a total of eighty-one analyzed models and a total of three hundred twenty-four simulations) as opposed to the other above FEA with a sample size of one (one patient and one model). The reduced number of FEA models originated in the difficulty of their creation (i.e., the higher the anatomical accuracy (high number of elements and nodes), the longer the time needed for the manual segmentation phase). The manual segmentation process supplies a better anatomical accuracy of the input data than the automated software segmentation, which is essential for accuracy of FEA results. Thus, here models had 28–184 times more elements and a global element size 10 times lower than the above FEA studies, supplying a better anatomical accuracy.

Another issue is related to the fact that the study here assessed the stress distribution over a model with a single tooth, without the adjacent teeth. Only one FEA study was found to compare multiple teeth vs. the one tooth model. Field et al. [29] reported for 0.5 N of tipping in intact periodontium a higher stress magnitude for the model with multiple teeth (324.5 KPa for PDL and 192.6 Kpa for dentin) vs. the model with a single tooth (235.5 Kpa for PDL and 87.6 Kpa for dentin). Nevertheless, both models [29] displayed amounts of stress exceeding by far the 16 Kpa of the MHP, suggesting ischemia, necrosis, and further loss which in clinical practice never occur. Thus, these [29] reported differences could not be verified. However, from the biomechanical point of view, a model with multiple teeth benefiting from the contact points should display lower amounts of stress than the model with a single tooth, since the adjacent teeth tend to limit the movements while the forces are dissipated through the contact points. However, further research is needed regarding this issue.

The assumed physical properties/conditions were isotropy, homogeneity, and linear elasticity, as in the above FEA studies, (despite the living tissues being non-homogenous, non-isotropic/anisotropic, and with non-linear elasticity). Biomechanically, under extremely small loads and movements the differences are particularly low among those two conditions. Additionally, the mathematical equations describing the anatomical reality implies extremely high computer power and is time consuming, with limited benefits.

Comparative studies of linearity vs. non-linearity conditions were performed only for PDL [26,27,28] subjected to intrusion. However, it must be emphasized that these three studies [26,27,28] employed idealized and simplified models of tooth and PDL, which could also influence the accuracy of the FEA results. Hemanth et al. [26,27] performed a simulation of the non-linearity vs. linearity in periodontal ligament under 0.2–0.3 N of intrusion, employing the brittle S1 and S3 material failure criteria (despite PDL being a ductile one). They showed that the quantitative difference is reduced by around 20% (less force is needed in non-linear equations compared with linear ones). Roscoe et al. [28] employed three failure criteria (von Mises—ductile, minimum principal S3—brittle, and hydrostatic stress—liquid) for the same non-linear vs. linear analysis of 0.25 N of intrusion in PDL, displaying an extremely unusual biomechanical color-coded stress distribution, for both type of properties and all criteria, and reporting that only hydrostatic stress in non-linear conditions is adequate, which contradicts all above FEA simulations as well as the analysis herein.

These analyses showed that each tooth and movement seems to display a particular localized area of maximum stress prone to resorptive risks (similar for the same tooth subjected to the same movement and independently to the amounts of applied force, which is in line with previous reports [1,2,3,4,5,6,7,17]). However, despite the individual variation of localized areas, there is a constant appearance of these prone to resorption areas in the same sector (apical, middle, cervical third) and/or root side for the analyzed teeth. Thus, despite Tresca (maximum shear stress) criteria supplying the localized areas of potential resorption risks, each individual tooth should be individually subject to FEA for identifying the particular areas prone to risks (seeming to be the subject of individual variation, in line with other reports [1,2,3,4,5,6,7,17]). This study supplyies the adequate criteria to be employed.

Despite these above limits and contradictory reports, the FEA remains the only available method to individually assess each component of an anatomical structure (and in particular the tooth and surrounding periodontium) [25]. However, there is a need for more corelated FEA with in vivo–in vitro studies aiming a further validation of the Tresca criteria (maximum shear stress) as better suited for the analysis of external root resorption.

## 5. Conclusions

Tresca failure criteria seem to be adequate for the FEA study of external root resorption and accurately supplies the color-coded localized areas of maximum shear stress prone to higher risks of resorptive process.The analysis of the radicular dentine–cementum component seems to provide a better accuracy of the areas prone to external root resorption than that of the entire tooth structure.The two light forces displayed similar extension of maximum stress areas.The localized resorption-prone areas seem to follow the progression of bone loss. Bone loss seems to indicate the risks of the resorptive process if the same applied force is guarded.The rotational and translational movements are prone to increase external resorptive risks, especially for 4–8 mm bone loss.The stress displayed in the coronal dentine decreases along with the progression of bone loss.The absorption–dissipation ability of the tooth is about 87.99–97.99% of the stress before reaching the periodontal ligament.

## 6. Practical Implications

For a practitioner, the identification of prone to external resorption localized areas is important when considering the orthodontic approach, especially if various bone loss levels are present. The rotational and translational movements with a higher potential risk of resorptive process in 4–8 mm reduced periodontium should be approached with care. Knowing the optimal interval of force (0.6–1.2 N) for all 5 movements and up to 8 mm of periodontal breakdown (known to be relatively safe for the dental pulp and periodontium) is important as well. To be able to know that from 60 gf (120 gf) only approx. 6 gf (12 gf) reach the periodontal ligament offers a better chance for orthodontic treatment success and reducing the ischemic and resorptive risks. Despite its difficulty and time-consuming nature, this method could also be clinically used if more data are necessary for potential resorptive risks of an individual tooth/teeth.

For the researchers, this study provides an adequate FEA failure criterion for the study of external root resorption, as well as providing a variety of results for a comparative study.

## Figures and Tables

**Figure 1 healthcare-11-01447-f001:**
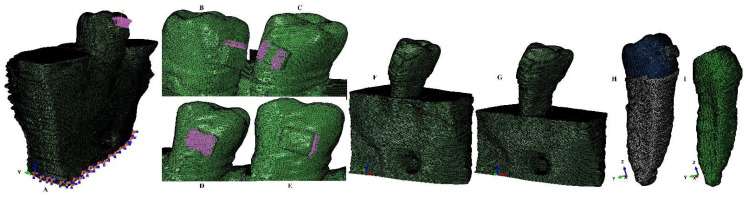
Mesh model: (**A**) second lower right premolar model with intact periodontium and applied vector for extrusion, (**B**) applied vector for intrusion, (**C**) applied vector for rotation, (**D**) applied vector for tipping, (**E**) applied vector for translation, (**F**) model with 4 mm bone loss, (**G**) model with 8 mm bone loss, (**H**) second mandibular premolar with intact PDL and applied bracket, (**I**) dentine component of the tooth.

**Figure 2 healthcare-11-01447-f002:**
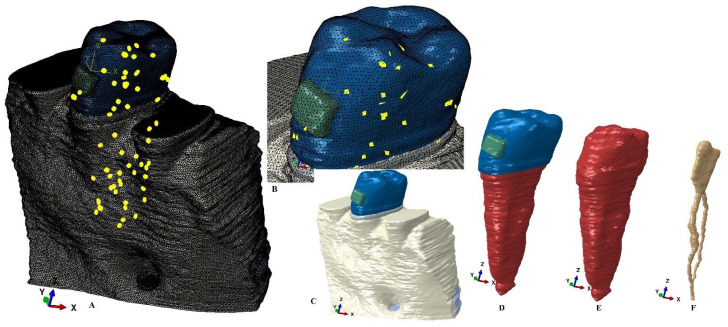
Mesh model: (**A**) model with intact periodontium and 63 elements warning for the tooth, bracket, and PDL, (**B**) 39 element warnings for tooth and bracket, (**C**) model with no bone loss without the mesh grid, (**D**) tooth with bracket and NVB without the mesh grid, (**E**) dentine component with NVB, (**F**) dental pulp with its NVB.

**Figure 3 healthcare-11-01447-f003:**
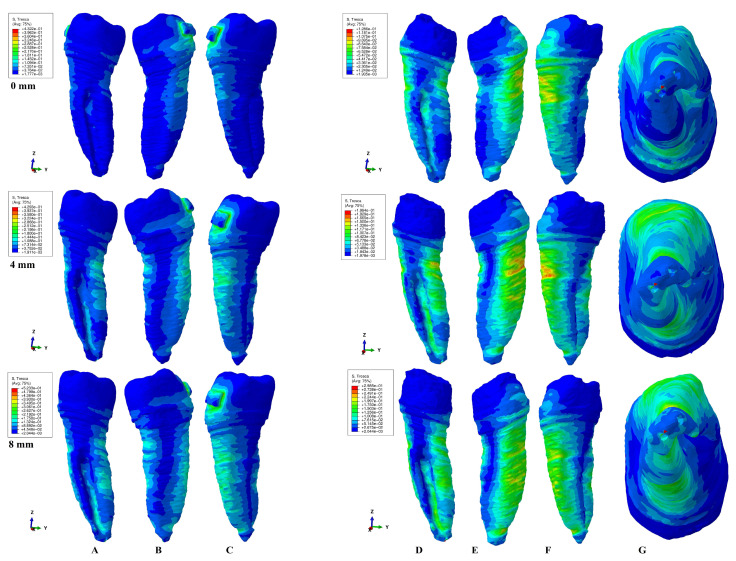
A total of 0.6 N of extrusion. Comparative shear stress display between the tooth structure in (**A**) lingual, (**B**) distal–vestibular, and (**C**) mesial–lingual view, and the dentine component in (**D**) lingual, (**E**) distal–vestibular, (**F**) mesial–lingual, and (**G**) apical view.

**Figure 4 healthcare-11-01447-f004:**
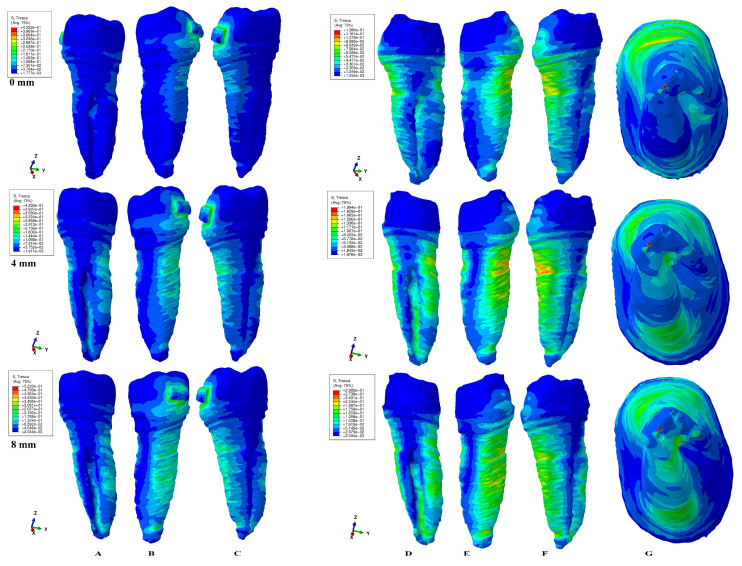
A total of 0.6 N of intrusion. Comparative shear stress display between the tooth structure in (**A**) lingual, (**B**) distal–vestibular, and (**C**) mesial–lingual view, and the dentine component in (**D**) lingual, (**E**) distal–vestibular, (**F**) mesial–lingual, and (**G**) apical view.

**Figure 5 healthcare-11-01447-f005:**
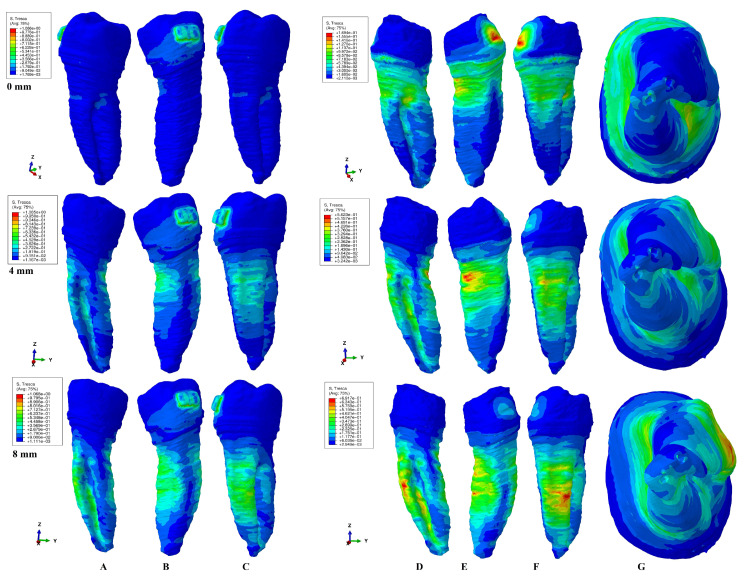
A total of 0.6 N of rotation. Comparative shear stress display between the tooth structure in (**A**) lingual, (**B**) distal–vestibular, and (**C**) mesial–lingual view, and the dentine component in (**D**) lingual, (**E**) distal–vestibular, (**F**) mesial–lingual, and (**G**) apical view.

**Figure 6 healthcare-11-01447-f006:**
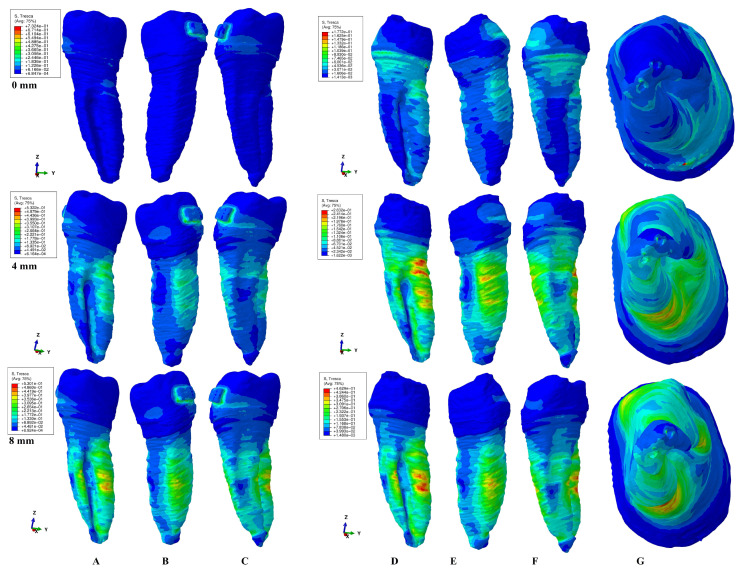
A total of 0.6 N of tipping. Comparative shear stress display between the tooth structure in (**A**) lingual, (**B**) distal–vestibular, and (**C**) mesial–lingual view, and the dentine component in (**D**) lingual, (**E**) distal–vestibular, (**F**) mesial–lingual, and (**G**) apical view.

**Figure 7 healthcare-11-01447-f007:**
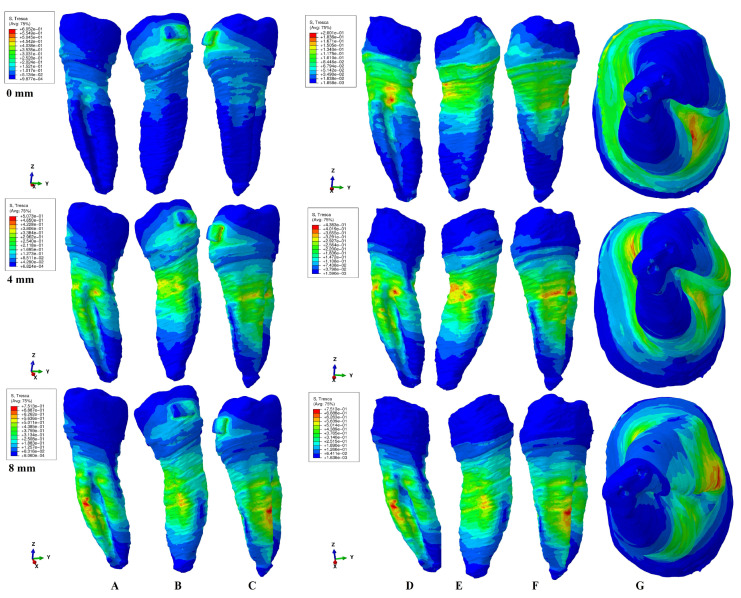
A total of 0.6 N of translation. Comparative shear stress display between the tooth structure in (**A**) lingual, (**B**) distal–vestibular, and (**C**) mesial–lingual view, and the dentine component in (**D**) lingual, (**E**) distal–vestibular, (**F**) mesial–lingual, and (**G**) apical view.

**Figure 8 healthcare-11-01447-f008:**
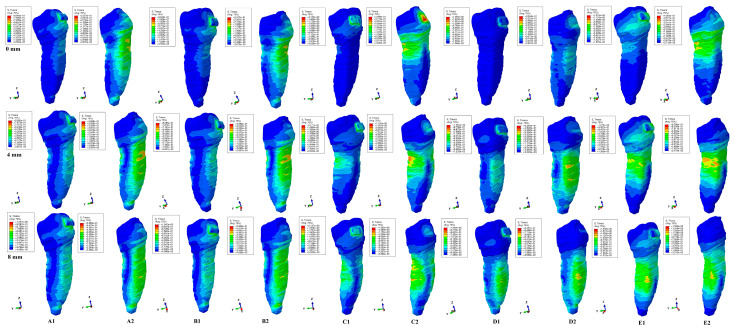
Comparative shear stress display for 1.2 N of intrusion ((**A1**) tooth, (**A2**) dentine), extrusion ((**B1**) tooth, (**B2**)), rotation ((**C1**) tooth, (**C2**) dentine), tipping ((**D1**) tooth, (**D2**) dentine), and translation ((**E1**) tooth, (**E2**) dentine).

**Table 1 healthcare-11-01447-t001:** Elastic properties of materials.

Material	Young’s Modulus, E (GPa)	Poisson Ratio, 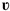	Refs.
Enamel	80	0.33	[8,9,10,11,12,18]
Dentin/Cementum	18.6	0.31	[8,9,10,11,12,18]
Pulp	0.0021	0.45	[8,9,10,11,12,18]
PDL	0.0667	0.49	[8,9,10,11,12,18]
Cortical bone	14.5	0.323	[8,9,10,11,12,18]
Trabecular bone	1.37	0.3	[8,9,10,11,12,18]
Bracket (Stainless Steel)	190	0.265	[8,9,10,11,12,18]

**Table 2 healthcare-11-01447-t002:** Maximum stress average values (KPa) produced by orthodontic forces in tooth structure and dentine.

Resorption (mm)			0	1	2	3	4	5	6	7	8
Intrusion	Structure	a	73.51	100.14	126.77	153.40	180.03	200.70	221.36	242.03	262.72
0.6 N/60 gf		m	73.51	100.14	126.77	153.40	180.03	200.70	221.36	242.03	262.72
		c	109.41	135.97	162.52	189.08	215.63	227.40	239.18	250.95	262.72
		C	145.20	162.81	180.42	198.02	215.63	227.40	239.18	250.95	262.72
	Dentine	a	44.18	74.76	105.34	135.92	166.50	187.15	207.81	228.46	249.11
		m	44.18	74.76	105.34	135.92	166.50	187.15	207.81	228.46	249.11
		c	107.50	130.48	153.45	176.43	199.40	211.83	224.26	236.68	249.11
		C	54.28	57.66	61.04	64.41	67.79	69.88	71.97	74.06	76.15
1.2 N/120 gf	Structure	a	147.02	200.28	253.54	306.80	360.05	401.39	442.72	484.05	525.44
		m	147.02	200.28	253.54	306.80	360.05	401.39	442.72	484.05	525.44
		c	218.82	271.93	325.04	378.15	431.25	454.80	478.35	501.90	525.44
		C	290.40	325.62	360.83	396.05	431.25	454.80	478.35	501.90	525.44
	Dentine	a	88.35	149.52	210.68	271.84	333.00	374.31	415.61	456.92	498.22
		m	88.35	149.52	210.68	271.84	333.00	374.31	415.61	456.92	498.22
		c	215.00	260.95	306.90	352.85	398.80	423.66	448.51	473.37	498.22
		C	108.56	115.32	122.07	128.83	135.57	139.76	143.94	148.12	152.30
Extrusion	Structure	a	73.51	100.14	126.77	153.40	180.03	200.70	221.36	242.03	262.72
0.6 N/60 gf		m	73.51	100.14	126.77	153.40	180.03	200.70	221.36	242.03	262.72
		c	109.41	135.97	162.52	189.08	215.63	227.40	239.18	250.95	262.72
		C	145.20	162.81	180.42	198.02	215.63	227.40	239.18	250.95	262.72
	Dentine	a	44.18	74.76	105.34	135.92	166.50	187.15	207.81	228.46	249.11
		m	44.18	74.76	105.34	135.92	166.50	187.15	207.81	228.46	249.11
		c	107.50	130.48	153.45	176.43	199.40	211.83	224.26	236.68	249.11
		C	54.28	57.66	61.04	64.41	67.79	69.88	71.97	74.06	76.15
1.2 N/120 gf	Structure	a	147.02	200.28	253.54	306.80	360.05	401.39	442.72	484.05	525.44
		m	147.02	200.28	253.54	306.80	360.05	401.39	442.72	484.05	525.44
		c	218.82	271.93	325.04	378.15	431.25	454.80	478.35	501.90	525.44
		C	290.40	325.62	360.83	396.05	431.25	454.80	478.35	501.90	525.44
	Dentine	a	88.35	149.52	210.68	271.84	333.00	374.31	415.61	456.92	498.22
		m	88.35	149.52	210.68	271.84	333.00	374.31	415.61	456.92	498.22
		c	215.00	260.95	306.90	352.85	398.80	423.66	448.51	473.37	498.22
		C	108.56	115.32	122.07	128.83	135.57	139.76	143.94	148.12	152.30
Translation	Structure	a	67.94	82.78	97.63	112.47	127.31	236.39	345.46	454.54	563.61
0.6 N/60 gf		m	67.94	156.66	245.38	334.10	465.02	536.60	608.19	679.77	751.35
		c	202.41	257.51	312.62	367.72	422.82	458.02	493.22	528.41	563.61
		C	202.42	236.42	270.41	304.41	338.40	363.43	388.47	413.50	438.53
	Dentine	a	67.94	78.67	89.40	100.12	110.85	208.50	306.16	403.81	501.46
		m	67.94	160.53	253.13	345.72	438.31	516.57	594.83	673.09	751.35
		c	200.11	250.38	300.65	350.91	401.18	441.88	482.57	523.27	563.96
		C	67.94	69.55	71.16	72.77	74.38	87.45	100.51	113.58	126.64
1.2 N/120 gf	Structure	a	135.88	165.57	195.25	224.94	254.61	472.77	690.92	909.07	1127.22
		m	135.88	313.32	490.76	668.20	930.05	1073.21	1216.37	1359.54	1502.69
		c	404.82	515.03	625.23	735.44	845.63	916.04	986.43	1056.83	1127.22
		C	404.83	472.83	540.82	608.81	676.80	726.87	776.93	827.00	877.05
	Dentine	a	135.88	157.34	178.79	200.25	221.69	417.01	612.31	807.62	1002.92
		m	135.88	321.07	506.25	691.44	876.62	1033.14	1189.66	1346.18	1502.69
		c	400.21	500.76	601.29	701.83	802.36	883.75	965.14	1046.53	1127.93
		C	135.88	139.10	142.32	145.54	148.76	174.89	201.02	227.15	253.28
Rotation	Structure	a	90.50	158.53	226.55	294.58	362.60	427.48	492.76	558.04	623.72
0.6 N/60 gf		m	90.50	181.11	271.72	362.32	452.93	540.10	627.27	714.44	801.61
		c	179.20	292.80	406.40	520.00	633.60	653.38	673.17	692.95	712.73
		C	356.62	358.12	359.61	361.11	362.60	383.41	404.22	425.03	445.84
	Dentine	a	71.84	136.25	200.66	265.07	329.48	376.99	424.50	472.01	519.52
		m	71.84	159.55	247.27	334.99	422.70	489.96	557.21	624.47	691.72
		c	169.46	267.67	365.89	464.10	562.31	565.96	569.62	573.27	576.92
		C	169.43	174.34	179.24	184.15	189.65	186.04	182.42	178.81	175.19
1.2 N/120 gf	Structure	a	181.00	317.05	453.10	589.15	725.20	854.96	985.52	1116.08	1247.45
		m	181.00	362.22	543.43	724.65	905.86	1080.20	1254.54	1428.88	1603.23
		c	358.40	585.60	812.80	1040.00	1267.20	1306.77	1346.33	1385.90	1425.46
		C	713.24	716.23	719.22	722.21	725.20	766.82	808.44	850.06	891.67
	Dentine	a	143.67	272.50	401.32	530.14	658.95	753.98	849.00	944.02	1039.04
		m	143.67	319.10	494.54	669.97	845.40	979.91	1114.42	1248.93	1383.44
		c	338.91	535.35	731.77	928.20	1124.63	1131.93	1139.23	1146.54	1153.84
		C	338.86	348.67	358.48	368.29	379.30	372.07	364.84	357.61	350.37
Tipping	Structure	a	61.67	90.71	119.74	148.78	177.81	232.79	287.76	342.74	397.71
0.6 N/60 gf		m	61.67	101.79	141.91	182.03	222.15	299.15	376.14	453.14	530.13
		c	122.61	169.65	216.68	263.72	310.75	332.49	354.23	375.97	397.71
		C	183.65	204.34	225.03	245.72	266.41	299.24	332.06	364.89	397.71
	Dentine	a	60.01	89.00	118.00	147.00	176.00	228.50	281.00	333.50	386.00
		m	60.01	99.92	139.84	179.75	219.66	280.48	341.29	402.11	462.92
		c	118.61	154.76	190.91	227.06	263.21	293.91	324.61	355.30	386.00
		C	60.01	61.76	63.51	65.26	67.01	69.85	72.70	75.54	78.38
1.2 N/120 gf	Structure	a	123.33	181.41	239.48	297.55	355.62	465.57	575.52	685.47	795.43
		m	123.33	203.58	283.82	364.06	444.30	598.29	752.28	906.27	1060.26
		c	245.23	339.29	433.36	527.43	621.50	664.98	708.46	751.94	795.43
		C	367.30	408.68	450.06	491.44	532.81	598.47	664.12	729.77	795.43
	Dentine	a	120.03	178.00	236.00	294.00	352.00	457.00	562.00	667.00	772.00
		m	120.03	199.85	279.67	359.50	439.33	560.95	682.58	804.21	925.83
		c	237.21	309.52	381.82	454.12	526.42	587.82	649.21	710.61	772.00
		C	120.03	123.52	127.02	130.52	134.02	139.71	145.39	151.08	156.77

Structure—stress displayed by the entire tooth structure. Dentine—stress displayed by the dentine component. a—root apical third, m—root middle third, c—root cervical third, C—crown.

**Table 3 healthcare-11-01447-t003:** Tresca criteria maximum stress average values (KPa) displayed in tooth structure, PDL, and % of the tooth quantitative stress that is displayed by the PDL (absorption–dissipation).

	Resorption (mm)		Apical	Middle	Cervical	% Apical	% Middle	% Cervical
Tooth	0	rotation	90.50	90.50	179.20	100.00	100.00	100.00
0.6 N/60 gf	8	rotation	623.72	801.61	712.73	100.00	100.00	100.00
	0	translation	67.94	67.94	202.41	100.00	100.00	100.00
	8	translation	563.61	751.35	563.61	100.00	100.00	100.00
	0	tipping	61.67	61.67	122.61	100.00	100.00	100.00
	8	tipping	397.71	530.13	397.71	100.00	100.00	100.00
	0	intrusion	73.51	73.51	109.41	100.00	100.00	100.00
	8	intrusion	262.72	262.72	262.72	100.00	100.00	100.00
	0	extrusion	73.51	73.51	109.41	100.00	100.00	100.00
	8	extrusion	262.72	262.72	262.72	100.00	100.00	100.00
1.2 N/120 gf	0	rotation	181.00	181.00	358.40	100.00	100.00	100.00
	8	rotation	1247.45	1603.23	1425.46	100.00	100.00	100.00
	0	translation	135.88	135.88	404.82	100.00	100.00	100.00
	8	translation	1127.22	1502.69	1127.22	100.00	100.00	100.00
	0	tipping	123.33	123.33	245.23	100.00	100.00	100.00
	8	tipping	795.43	1060.26	795.43	100.00	100.00	100.00
	0	intrusion	147.02	147.02	218.82	100.00	100.00	100.00
	8	intrusion	525.44	525.44	525.44	100.00	100.00	100.00
	0	extrusion	147.02	147.02	218.82	100.00	100.00	100.00
	8	extrusion	525.44	525.44	525.44	100.00	100.00	100.00
PDL	0	rotation	2.34	4.62	18.35	2.59	5.11	10.24
Ref. [11]	8	rotation	4.62	19.62	68.28	0.74	2.45	9.58
0.6 N/60 gf	0	translation	2.10	3.97	17.71	3.09	5.84	8.75
	8	translation	7.51	15.10	67.70	1.33	2.01	12.01
	0	tipping	1.55	3.06	12.13	2.51	4.96	9.89
	8	tipping	8.26	12.34	44.91	2.08	2.33	11.29
	0	intrusion	3.00	3.00	5.22	4.08	4.08	4.77
	8	intrusion	8.25	8.25	18.43	3.14	3.14	7.02
	0	extrusion	3.00	3.00	6.70	4.08	4.08	6.12
	8	extrusion	8.25	8.25	22.55	3.14	3.14	8.58
1.2 N/120 gf	0	rotation	4.67	9.25	36.70	2.58	5.11	10.24
	8	rotation	19.79	39.24	136.56	1.59	2.45	9.58
	0	translation	4.03	7.95	35.43	2.97	5.85	8.75
	8	translation	15.02	30.20	135.40	1.33	2.01	12.01
	0	tipping	3.10	6.12	24.26	2.51	4.96	9.89
	8	tipping	16.53	24.68	89.02	2.08	2.33	11.19
	0	intrusion	5.99	5.99	10.44	4.07	4.07	4.77
	8	intrusion	16.50	16.50	36.86	3.14	3.14	7.02
	0	extrusion	5.99	5.99	13.41	4.07	4.07	6.13
	8	extrusion	16.50	16.50	45.10	3.14	3.14	8.58

Apical—apical third, middle—middle third, cervical—cervical third. % apical—% apical third, % middle—% middle third, % cervical—% cervical third. tooth—entire tooth structure.

## Data Availability

Not applicable.

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
