# Peer review of "Assessment of the Orthodontic External Resorption in Periodontal Breakdown—A Finite Elements Analysis (Part I)"

_healthcare, 2023, doi:10.3390/healthcare11101447_

Round 1
Reviewer 1 Report
Well done and huge work. Well described and organised. Some sentence should be imoroved. Example in line 679, first sentences of Conclusion. Before that short description should be done.
Well done and huge work. Well described and organised. Some sentence should be imoroved. Example in line 679, first sentences of Conclusion. Before that short description should be done.
Author Response
Corresponding author
Department of Cariology, Endodontics and Oral Pathology
Faculty of Dental Medicine
University of Medicine and Pharmacy
Ms. Nora-Evelyn Vrannai
Assistant Editor
Healthcare
Special Issue - Second Edition of Innovative Solutions for Oral Healthcare
May 13, 2023
Dear Ms. Nora-Evelyn Vrannai,
Thank you very much for your letter dated May 12, 2023, with the comments of the reviewers. We have now carefully considered the comments of the reviewers and amended the paper accordingly. All changes are highlighted in red throughout the manuscript and included also below.
Reply to Reviewer #1:
We agree and we thank the reviewer for his/her time and comments. Appropriate changes in the manuscript have by now been made. Please see below and in the manuscript.
Concern of the reviewer:
”Comments and Suggestions for Authors
Well done and huge work. Well described and organised. Some sentence should be improved. Example in line 679, first sentences of Conclusion. Before that short description should be done.
Comments on the Quality of English Language
Well done and huge work. Well described and organised. Some sentence should be improved. Example in line 679, first sentences of Conclusion. Before that short description should be done.”
Point-by-point response to the reviewer’s comments:
Our response:
- We thank the reviewer for his/her concern and comments. We do hope that our changes are according to the reviewer‘s remarks.
Revised text: The recommended corrections were done. Please see the entire manuscript.
The conclusions section revised text (pg.22, lines 710-722):
“1. Tresca failure criteria seem to be adequate for the FEA study of external root resorption and accurately supplies the color-coded localized areas of maximum shear stress prone to higher risks of resorptive process.
- The analysis of the radicular dentine-cementum component seems to provide a better accuracy of the areas prone to external root resorption than that of the entire tooth structure.
3.The two light forces displayed similar extension of maximum stress areas.
4.The localized resorption prone areas seem to follow the progression of bone loss. Bone loss seems to favorize the risks of resorptive process if the same applied force is guarded.
5.The rotational and translational movements are prone to increase external resorptive risks especially for 4-8 mm bone loss.”

Reviewer 2 Report
The scientific article is a little bit too long, I suggest to reduce the discussion and even maybe the table and figures, if possible.
Author Response
Corresponding author
Department of Cariology, Endodontics and Oral Pathology
Faculty of Dental Medicine
University of Medicine and Pharmacy
Ms. Nora-Evelyn Vrannai
Assistant Editor
Healthcare
Special Issue - Second Edition of Innovative Solutions for Oral Healthcare
May 13, 2023
Dear Ms. Nora-Evelyn Vrannai,
Thank you very much for your letter dated May 12, 2023, with the comments of the reviewers. We have now carefully considered the comments of the reviewers and amended the paper accordingly. All changes are highlighted in red throughout the manuscript and included also below.
Reply to Reviewer #2:
We agree and we thank the reviewer for his/her time and comments.
Concern of the reviewer:
”Comments and Suggestions for Authors
The scientific article is a little bit too long, I suggest to reduce the discussion and even maybe the table and figures, if possible.”
Point-by-point response to the reviewer’s comments:
Our response:
- We thank the reviewer for his/her concern and comments. We agree that the article is a little bit too long. However, being the first research of this type a detailed approach of the various aspects of the finite analysis of the root resorption helps the reader (both the practitioner and the researcher) to better understand all these issues.

Reviewer 3 Report
In this finite elements analysis the authors investigated the risk of external root resorption at mandibular second premolars following orthodontic tooth movement at different horizontal levels of the surrounding periodontal tissue.
ABSTRACT
- Point out the clinical conclusions in the abstract too.
METHODS
- What was the justifying indication of the CBCTs taken?
- Which settings were used for the CBCT?
RESULTS, FIGURES, TABLES
- Table 1 requires a table legend.
- Figure 3-7: increase the size of the labelling.
DISCUSSION
- Please shorten the discussion section.
LANGUAGE
- English language (typing errors, grammar, and syntax) needs to be revised thoroughly throughout the text.
The study addresses an interesting research question, but some corrections of the text are required.
Therefore, I suggest to accept this study with minor revision being necessary.
The syntax of the sentences needs to be revised (e.g. "The excessive orthodontic amount of force").
Please check that singular and plural are correctly used (e.g. "with an extremely variable positions").
Please check the grammar (e.g. "did not interfered").
Author Response
Corresponding author
Department of Cariology, Endodontics and Oral Pathology
Faculty of Dental Medicine
University of Medicine and Pharmacy
Ms. Nora-Evelyn Vrannai
Assistant Editor
Healthcare
Special Issue - Second Edition of Innovative Solutions for Oral Healthcare
May 14, 2023
Dear Ms. Nora-Evelyn Vrannai,
Thank you very much for your letter dated May 12, 2023, with the comments of the reviewers. We have now carefully considered the comments of the reviewers and amended the paper accordingly. All changes are highlighted in red throughout the manuscript and included also below.
Reply to Reviewer #3:
We agree and we thank the reviewer for his/her time and comments. Appropriate changes in the manuscript have by now been made. Please see below and in the manuscript.
Concern of the reviewer:
”Comments and Suggestions for Authors
In this finite elements analysis the authors investigated the risk of external root resorption at mandibular second premolars following orthodontic tooth movement at different horizontal levels of the surrounding periodontal tissue.
ABSTRACT
- Point out the clinical conclusions in the abstract too.
METHODS
- What was the justifying indication of the CBCTs taken?
- Which settings were used for the CBCT?
RESULTS, FIGURES, TABLES
- Table 1 requires a table legend.
- Figure 3-7: increase the size of the labelling.
DISCUSSION
- Please shorten the discussion section.
LANGUAGE
- English language (typing errors, grammar, and syntax) needs to be revised thoroughly throughout the text.
The study addresses an interesting research question, but some corrections of the text are required.
Therefore, I suggest to accept this study with minor revision being necessary.
Comments on the Quality of English Language
The syntax of the sentences needs to be revised (e.g. "The excessive orthodontic amount of force").
Please check that singular and plural are correctly used (e.g. "with an extremely variable positions").
Please check the grammar (e.g. "did not interfered").”
Point-by-point response to the reviewer’s comments:
- Concern of the reviewer:
“ABSTRACT
- Point out the clinical conclusions in the abstract too.”
Our response:
- We thank the reviewer for his/her concern and comments. We do hope that our changes are according to the reviewer‘s remarks.
Revised text: pg.1 lines15-30
“…Tresca criteria accurately displayed the localized areas of maximum stress prone to external resorption risks, seeming to be adequate for the study of resorptive process. The localized areas were better displayed in the radicular dentine-cementum component than in the entire tooth structure. The rotation and translation seem prone to a higher risk of external root resorption after 4 mm of loss. The resorptive risks seem to increase along with the progression of periodontal breakdown if the same amount of applied force is guarded. The localized resorption prone areas follow the progression of bone loss. The two light forces displayed similar extension of maximum stress areas. The stress displayed in the coronal dentine decreases along with the progression of bone loss. The absorption-dissipation ability of the tooth is about 87.99-97.99% of the stress.”
- Concern of the reviewer:
“METHODS
- What was the justifying indication of the CBCTs taken?
- Which settings were used for the CBCT?.”
Our response:
- We thank the reviewer for his/her concern and comments. We do hope that our changes are according to the reviewer‘s remarks.
Revised text: pg.4 lines 201-205
“The initial information on which our 3D models were based came from radiological analysis performed for diagnostic and control purposes using CBCT (cone beam computed tomography) examination (of the mandibular arch with the two premolars and molars), with a voxel size of 0.075 mm, FOV 50 x 50 mm (ProMax 3DS, Planmeca, Helsinki, Finland).”
- Concern of the reviewer:
“RESULTS, FIGURES, TABLES
- Table 1 requires a table legend.
- Figure 3-7: increase the size of the labelling.”
Our response:
- We thank the reviewer for his/her concern and comments. We do hope that our changes are according to the reviewer‘s remarks.
Revised text: pg. 5 line 229
“Table 1 Elastic properties of materials”
Changed figures will be provided to the editor.
- Concern of the reviewer:
“DISCUSSION
Please shorten the discussion section.”
Our response:
- We thank the reviewer for his/her concern and comments. We agree that the article is a little bit too long. However, being the first research of this type a detailed approach of the various aspects of the finite analysis of the root resorption helps the reader (both the practitioner and the researcher) to better understand all these issues.
- Concern of the reviewer:
“LANGUAGE
- English language (typing errors, grammar, and syntax) needs to be revised thoroughly throughout the text.
The study addresses an interesting research question, but some corrections of the text are required.
Therefore, I suggest to accept this study with minor revision being necessary.
Comments on the Quality of English Language
The syntax of the sentences needs to be revised (e.g. "The excessive orthodontic amount of force").
Please check that singular and plural are correctly used (e.g. "with an extremely variable positions").
Please check the grammar (e.g. "did not interfered").”
Our response:
- We thank the reviewer for his/her concern and comments. We do hope that our changes are according to the reviewer‘s remarks.
Revised text: entire manuscript

Reviewer 4 Report
I would like to congratulate all the authors for the interesting and extensive work they have been developing, in a subject that affects so many orthodontists with complications that are difficult to control and solve.
I will make some comments and suggestions in order to improve the manuscript:
-In the introduction, the authors made an exhaustive review of the literature with actual and important bibliography. I think it would be useful to clarify if there is evidence that orthodontic re-treatments are more susceptible to induce radicular resorption.
-The topic of resorptions, although not new, in your work is studied through FEA, is your study different from others already carried out? Does it address a specific gap in the field?
-Concerning to methodology, I would like to know and to be added what was the criteria of choosing the premolar teeth for the study.
-Discussion is well done according to the results and supported by appropriate references.
-Conclusions are well understood and clear I would like you to enumerate objectives (main and secondary) in accordance with the conclusions presented.
Figure 1 the letters A-I are not very clear.
Author Response
Corresponding author
Department of Cariology, Endodontics and Oral Pathology
Faculty of Dental Medicine
University of Medicine and Pharmacy
Ms. Nora-Evelyn Vrannai
Assistant Editor
Healthcare
Special Issue - Second Edition of Innovative Solutions for Oral Healthcare
May 14, 2023
Dear Ms. Nora-Evelyn Vrannai,
Thank you very much for your letter dated May 12, 2023, with the comments of the reviewers. We have now carefully considered the comments of the reviewers and amended the paper accordingly. All changes are highlighted in red throughout the manuscript and included also below.
Reply to Reviewer #4:
We agree and we thank the reviewer for his/her time and comments. Appropriate changes in the manuscript have by now been made. Please see below and in the manuscript.
Concern of the reviewer:
” I would like to congratulate all the authors for the interesting and extensive work they have been developing, in a subject that affects so many orthodontists with complications that are difficult to control and solve.
I will make some comments and suggestions in order to improve the manuscript:
-In the introduction, the authors made an exhaustive review of the literature with actual and important bibliography. I think it would be useful to clarify if there is evidence that orthodontic re-treatments are more susceptible to induce radicular resorption.
-The topic of resorptions, although not new, in your work is studied through FEA, is your study different from others already carried out? Does it address a specific gap in the field?
-Concerning to methodology, I would like to know and to be added what was the criteria of choosing the premolar teeth for the study.
-Discussion is well done according to the results and supported by appropriate references.
-Conclusions are well understood and clear I would like you to enumerate objectives (main and secondary) in accordance with the conclusions presented.
Figure 1 the letters A-I are not very clear.”
Point-by-point response to the reviewer’s comments:
- Concern of the reviewer:
“-In the introduction, the authors made an exhaustive review of the literature with actual and important bibliography. I think it would be useful to clarify if there is evidence that orthodontic re-treatments are more susceptible to induce radicular resorption.
-The topic of resorptions, although not new, in your work is studied through FEA, is your study different from others already carried out? Does it address a specific gap in the field?”
Our response:
- We thank the reviewer for his/her concern and comments. We do hope that our changes are according to the reviewer‘s remarks.
Revised text: pg.2 lines56-57
“There are no available studies about the issue of orthodontic re-treatments and the radicular resorption risks.”
pg.4 lines166-170
“This research is the first of this type to study the correlations of the external root resorption and the horizontal periodontal breakdown, employing an adequate failure criterion (suited and documented for these tissues) which supplies correct results. Earlier FEA studies by not using the adequate failure criteria failed in supplying exact location of areas susceptible to resorptive processes.”
- Concern of the reviewer:
“-Concerning to methodology, I would like to know and to be added what was the criteria of choosing the premolar teeth for the study.”
Our response:
- We thank the reviewer for his/her concern and comments. We do hope that our changes are according to the reviewer‘s remarks.
Revised text: pg.4 lines 194-196
“The mandibular premolars were selected both out of a desire to study the area with higher loadings and periodontal problems and because of the finite element studies available in the literature to perform data comparison and validation.”
- Concern of the reviewer:
“-Discussion is well done according to the results and supported by appropriate references.
-Conclusions are well understood and clear I would like you to enumerate objectives (main and secondary) in accordance with the conclusions presented.
Figure 1 the letters A-I are not very clear.”
Our response:
- We thank the reviewer for his/her concern and comments. We do hope that our changes are according to the reviewer‘s remarks.
Revised text: pg.4 lines 171-176
“The aims of the present simulation were: a) assessment of Tresca criteria as adequate for the study of root resorption, b) the assessment of the areas more prone to orthodontic external root resorption when 0.6 N/60 gf and 1.2 N/120 gf are applied during the 0-8 mm periodontal breakdown process under multiple orthodontic forces, c) the ability of dental structures to absorb and dissipate the two applied forces.”
Changed figures will be provided to the editor.

Reviewer 5 Report
Dear authors!
The state of this research is to examine how different orthodontic forces applied to teeth impact their stress-strain state, while also taking into consideration bone resorption. The authors' approach in studying this issue is distinctive as a set of mathematical models were developed to form a sample size. Indeed, such studies are extremely rare due to the difficulty of preparing mathematical models. The data gathered and presented by the authors has significant theoretical relevance as it elucidates the biomechanical characteristics that are important for predicting the success of orthodontic treatment.
In reviewing the research material, I highlighted some suggestions that may be helpful:
1. Title. The term "Risks Assessment" was used in the title, which, in my opinion, is not quite correct, because risk assessment involves a method of statistical analysis, the result of which should be expressed in the probability of certain event occurrence. Indeed, the risks can be evaluated based on the study data, however these judgments can only be indirect.
2. Please clarify the absence of adjacent teeth. Could the presence of adjacent teeth beside the premolar under investigation affect the stress distribution in your model? Did you consider the absence of contact with adjacent teeth and the load and stress distribution to shift through the contact points? Is this a limitation of the study?
Author Response
Corresponding author
Department of Cariology, Endodontics and Oral Pathology
Faculty of Dental Medicine
University of Medicine and Pharmacy
Ms. Nora-Evelyn Vrannai
Assistant Editor
Healthcare
Special Issue - Second Edition of Innovative Solutions for Oral Healthcare
May 15, 2023
Dear Ms. Nora-Evelyn Vrannai,
Thank you very much for your letter dated May 12, 2023, with the comments of the reviewers. We have now carefully considered the comments of the reviewers and amended the paper accordingly. All changes are highlighted in red throughout the manuscript and included also below.
Reply to Reviewer #5:
We agree and we thank the reviewer for his/her time and comments. Appropriate changes in the manuscript have by now been made. Please see below and in the manuscript.
Concern of the reviewer:
” Comments and Suggestions for Authors
Dear authors!
The state of this research is to examine how different orthodontic forces applied to teeth impact their stress-strain state, while also taking into consideration bone resorption. The authors' approach in studying this issue is distinctive as a set of mathematical models were developed to form a sample size. Indeed, such studies are extremely rare due to the difficulty of preparing mathematical models. The data gathered and presented by the authors has significant theoretical relevance as it elucidates the biomechanical characteristics that are important for predicting the success of orthodontic treatment.
In reviewing the research material, I highlighted some suggestions that may be helpful:
- Title. The term "Risks Assessment" was used in the title, which, in my opinion, is not quite correct, because risk assessment involves a method of statistical analysis, the result of which should be expressed in the probability of certain event occurrence. Indeed, the risks can be evaluated based on the study data, however these judgments can only be indirect.
- Please clarify the absence of adjacent teeth. Could the presence of adjacent teeth beside the premolar under investigation affect the stress distribution in your model? Did you consider the absence of contact with adjacent teeth and the load and stress distribution to shift through the contact points? Is this a limitation of the study?”
Point-by-point response to the reviewer’s comments:
- Concern of the reviewer:
“-Title. The term "Risks Assessment" was used in the title, which, in my opinion, is not quite correct, because risk assessment involves a method of statistical analysis, the result of which should be expressed in the probability of certain event occurrence. Indeed, the risks can be evaluated based on the study data, however these judgments can only be indirect.”
Our response:
- We thank the reviewer for his/her concern and comments. We do hope that our changes are according to the reviewer‘s remarks.
Revised text: pg.1 line 2
- Concern of the reviewer:
“Please clarify the absence of adjacent teeth. Could the presence of adjacent teeth beside the premolar under investigation affect the stress distribution in your model? Did you consider the absence of contact with adjacent teeth and the load and stress distribution to shift through the contact points? Is this a limitation of the study.”
Our response:
- We thank the reviewer for his/her concern and comments. We do hope that our changes are according to the reviewer‘s remarks.
Revised text: pg.21 lines 659-671
“Another issue is related to the fact that the study here assessed the stress distribution over a model with a single tooth, without the adjacent teeth. Only one FEA study was found to compare multiple teeth vs. one tooth model. Field et al. [29] reported for 0.5 N of tipping in intact periodontium a higher stress magnitude for the model with multiple teeth (324.5 KPa for PDL and 192.6 KPa for dentin) vs. the model with a single tooth (235.5 KPa for PDL and 87.6 KPa for dentin). Nevertheless, both models [29] displayed amounts of stress exceeding by far the 16 KPa of the MHP, suggesting ischemia, necrosis, and fur-ther loss which in clinical practice never occur. Thus, these [29] reported differences could not be verified. However, from the biomechanical point of view, a model with multiple teeth benefiting from the contact points should display lower amounts of stress than the model with a single tooth since the adjacent teeth tend to limit the movements while the forces are dissipated through the contact points. However, further research is needed into this issue.”
